# Novel Procedures for Sustainable Design in Structural Rehabilitation on Oversized Metal Structures

Ștefan Mocanu [1], Laurențiu Rece [1], Adrian Burlacu [2,*], Virgil Florescu [1], Corneliu Ronțescu [3] and Arina Modrea [4]

1 Mechanical Technology Department, Technological Equipment Faculty, Technical University of Civil Engineering of Bucharest, 020396 Bucharest, Romania; stefan.mocanu@utcb.ro (Ș.M.); rece@utcb.ro (L.R.); virgil.florescu@utcb.ro (V.F.)
2 Department of Roads, Railways and Construction Materials, Faculty of Railways, Roads and Bridges, Technical University of Civil Engineering of Bucharest, 020396 Bucharest, Romania
3 Quality Engineering and Industrial Technologies Department, Faculty of Industrial Engineering and Robotics, University Politehnica of Bucharest, 060042 Bucharest, Romania; corneliu.rontescu@yahoo.com
4 Pharmacy, Science and Technology George Emil Palade Targu Mures, University of Medicine, 300134 Targu Mures, Romania; arina.modrea@umfst.ro
* Correspondence: adrian.burlacu@utcb.ro; Tel.: +40-723-645-722

**Abstract:** This article includes several studies and advanced research carried out on the subject of large KANGUR-type port cranes, subjected to regular or interventional maintenance work; besides the beneficial effects for which they were intended, some undesirable effects were detected in terms of affecting safety in exploitation. The purpose of this paper is to identify the risk factors and to offer sustainable solutions for increasing operational safety and service life for the respective equipment, as well as for other similar fields of activity, including oversized structures. Following the theoretical studies and the experimental research carried out in situ that confirmed the validity of the theoretical models used, the article in conclusion provides specific and sustainable solutions as well as new methods of assessment, which combine the positive elements of the usual solutions, but in parallel, allow the elimination of the negative side effects that may occur in time. Hence, the article's novelties also consist of a new approach to achieve structural interventions, resulting in the increase in the service life, and in parallel with that, the safety in operation of the respective equipment.

**Keywords:** failure mode; elastic instability; critical force; buckling critical force; FEM (finite-element method)

## 1. Introduction

KANGUR 2-type equipment is a port-lifting facility of large-size portico-crane type. The crane is considered critical equipment, with a high degree of risk in operation and being subject to specific national and international regulations. Periodic tests and safety procedures are provided, including overload tests, in order to ensure their safe operation. The problem becomes even more complex when it is necessary to assess the installation's remaining lifespan according to the existing regulations.

The research was based on the application of the finite-element method and related simulation capabilities, followed by certain analytical results in order to provide model validation by means of tensometric measurements on the physical structure (on identical equipment).

Extensive studies on the subject of mechanical performance of cranes, as well as deep analysis based on operational experience in the field across long timeframes, enabled the creation of predefined engineering packages and procedures, as well as well-constructed international standards meant to assure quality, safety, environmental safety and structural soundness in normal operation as well as special circumstances (e.g., earthquakes).

Both OSHA (Occupational Safety and Health Administration) and ASME (American Society of Mechanical Engineering), as well as the Romanian equivalent ISCIR (Inspecția de Stat pentru Controlul Cazanelor, Recipientelor sub Presiune și Instalațiilor de Ridicat) provide detailed requirements and descriptions on the subject of regulating cranes both in terms of design and manufacturing as well as in operation. OSHA 1918.51 provides limits and qualitative requirements regarding operating conditions expected in the use of maritime-port cranes, along with the 1918.55 standard that gives specific legal limits in terms of operating conditions. ASME B30 provides guidelines for the design and safety requirements imposed on producers of cranes as well as the material, treatment and inspection conditions underwent by the technical asset. The aforementioned pieces of legislation were considered for the design considerations and limitations imposed by the international legal framework for the design and analysis tasks hereby provided. Depale et al. [1] provide an overview and analysis for FEM Crane Codes as well as the EN European Standards for steel structures and cranes.

Lee et al. [2] provide extensive structural-life considerations, including an in-depth assessment covering structural fatigue, environmental impact, life estimates, maintenance as well as economic factors. The structural models presented within this study formed the basis for understanding the limit and ultimate conditions taken into consideration in the design, analysis and testing of cranes. Corigliano et al. [3] further extend the study, exploring the specific properties of structural steel on the structural (tensile, thermoresidual and fatigue) properties of cranes, which provided great aid in understanding validation of the theoretical formulae and understanding with real-life data.

Marquez et al. [4] provide a great wealth of knowledge compiled into a literature review consisting of common causes for failure of cranes, as well as their root-cause-analysis reports. This paper was used in the development of the mental framework, in guiding design decisions and developing the analysis plan in accordance with the common failure causes discovered.

Buczkowski et al. [5] provide a guiding example of an FEA analysis developed to ascertain mechanical and structural performance of crane structures, whilst Sakhvidi et al. [6] focus on Shell elements and vibrations. Foraboschi, P. [7] introduce lateral loads in the analysis, thus providing insight into this loading case. Gusella et al. [8] tackle the issue of ductility as an important mechanical property for this analysis with regards to the redistribution of moment loading.

Kang et al. [9] studied the influence of the thickness of a collar-stiffening element around a wall opening on the final strength. The results were compared with the initial design criteria, and using a regression analysis, proposed an efficient equation for determining the thickness of the collar stiffness for a large-diameter cylindrical shell. Several studies examined the stress–strain distribution in different type of shells with curvilinear openings. Guz et al. [10], Zirka et al. [11] and Alsalah. et al. [12] analyzed the effect on the opening of a cylindrical shell by conducting stress-concentration tests. A Molecular Dynamics (MD) Simulation analysis for the metallic alloy structure, as proposed by Dongpeng Hua et al. [13], will be the subject of further research.

Recently acquired good practice and novel methods used by the above-mentioned authors represent a valid starting point concerning the present work approach. The article novelty consists in highlighting metallic-structure failure as a result of the real consequences of some solutions currently adopted in the maintenance activity for this equipment presenting risk in operation. Starting from the currently used methods of intervention, in the first stage, a theoretical study is made using the finite-element method, concerning the efficiency of these types of interventions in terms of operational safety, hence reducing the risk of major accidents. Sustainable solutions for strengthening/repairing the main structural-frame elements (affected in time as a result of exploitation and the influence of environmental factors) are offered here.

In a second stage, a complex analysis is conducted, through experimental research, which is based on the behavior in operation; in particular, of the worn structures that have

been subjected to rehabilitation. The aim is to identify the main causes that led to the failure of some interventions on the structure (accidents, malfunctions, etc.) as well as on the appropriate countermeasures. The study involves a 1:1 scale modeling of the structural elements, followed by a simulation, the results of which were validated in practice by experimental measurements in situ.

## 2. Materials and Methods

The study was carried out by going through the following main steps:

- real scale (1:1) 3D model using SolidWorks software, 2014, Dassault Systèmes, France;
- performing a static study simulation of the structure behavior;
- performing tensometric measurements in order to describe the strain–stress field under load;
- conducting an analysis regarding the fatigue behavior of the arm with the help of SolidWorks Simulation suite [14–16].

The first stage consists of kinematic-structure-element 3D modeling, starting from original archive drawings (Figures S1–S5, Annex 1), by means of SolidWorks software suite. General good-practice modeling recommendations such as fine-detail modeling without unneeded parameters were taken into account in order to avoid cumbersome meshing schemes, which are sometimes hard or impossible to use.

Some of the fundamental elements of modeling worthy of mention include the use of as few "loft"-type steps as possible for the materialization of geometric-pattern characteristics along the longitudinal axis of a given element; the use at the level of solid body (and not sketch—sketch) of the "fillet" type commands—the fillet radius or "chamfer"—the bevel; and avoidance for unneeded details that do not contribute in a defining way to the study stage (the handling ears of the elements, auxiliary constructions such as bridges, railings, stairs, etc.), their gravitational effect being subsequently taken into account by loads distributed along the length of the element. By modeling all the components, the entire structure of the lifting system of the crane could be achieved, respectively the model of the main boom (Figure 1).

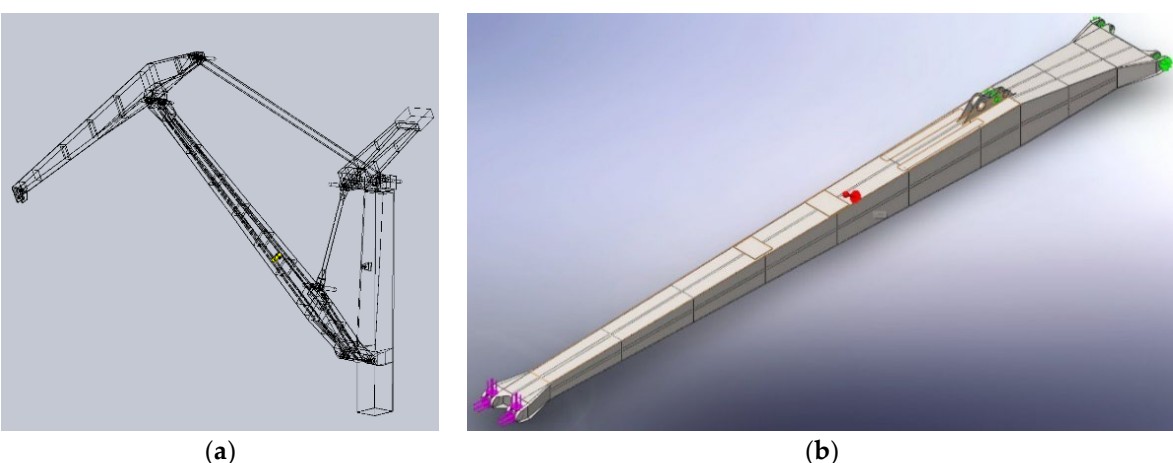

(**a**)  (**b**)

**Figure 1.** (**a**) KANGUR crane assembly, (**b**) main arm, 1:1 scale modeling.

Despite the general requirements for maximum simplification of the elements modeled by the above-mentioned measures, the modeling took into account in detail the original design of the studied components, highlighting any element that could influence the final structural behavior under load. In this respect, there were studied two constructive variants of the main arm of the equipment (Figure 2), which have as reference the sets of drawings existing within the technical documentation. The similarity of the exterior geometrical construction is noted (for understandable reasons), the interior construction being distinguished by a completely redesigned shape of the diaphragms as well as another

way of solidarizing the strengthening rails of the "flange-type" and "web-type" elements, practically obtaining a new structure.

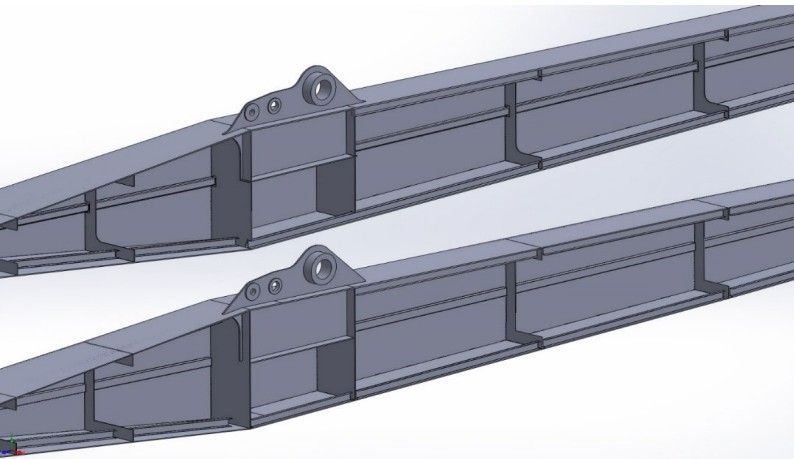

**Figure 2.** Details of modeling for the internal structure of the boom.

The holder performed interventions on the boom of the crane that was broken, materialized by the plating of its superior flange-type elements. The plating scheme (repair/consolidation) of the boom was taken by in situ survey and from the drawing that completes the repair documentation. The method of plating is shown in Figure 3.

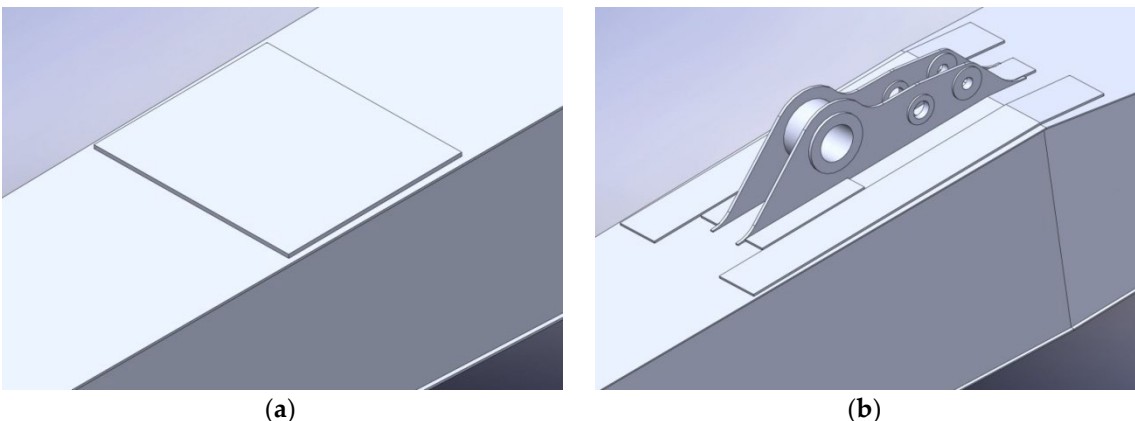

| (**a**) | (**b**) |
|---|---|

**Figure 3.** Plated arm (**a**) middle arm section (**b**) middle hinge section.

A kinematic study was conducted. In order to complete the kinematic study stage, the actual ensemble of the lifting equipment was modeled as a SolidWorks "assembly" document having as components the previously modeled subassemblies (Figure 4). The output data for this study consist of the relative and absolute angles of the subassemblies according to the general operating parameters provided by the original documentation. The static equilibrium condition of the assembly is satisfied by modeling the length rack of the operating motor of the corresponding boom opening (working position) or by placing three-axis hinged supports at the required points (Figure 4).

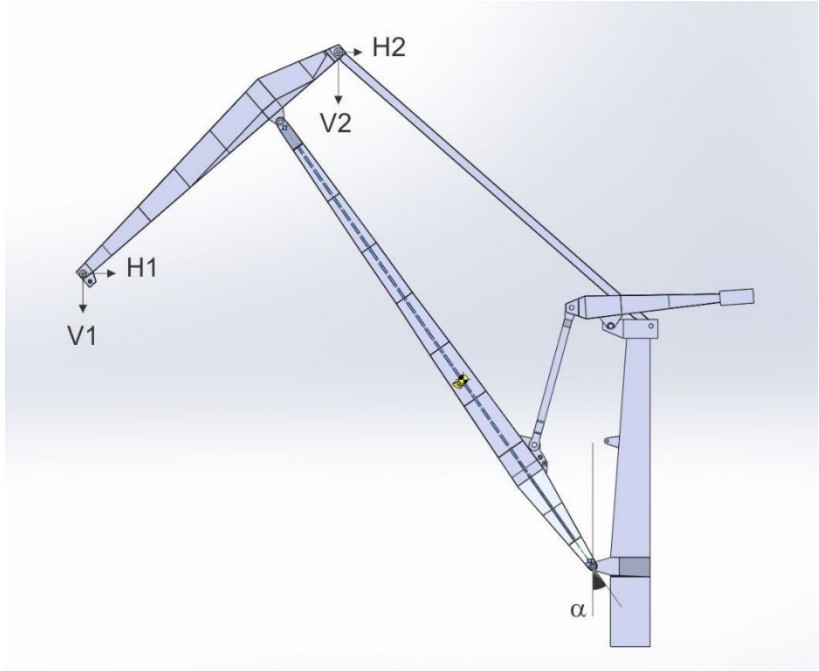

**Figure 4.** Variables that define the study.

The two fundamental variables that characterize each study are the angle of inclination of the main section to the vertical (alpha) and the considered workload.

Thus, iterations were completed for various positions and/or work tasks, the purpose being to declare a so-called disadvantageous case, one of the chosen performance criteria being the tension and/or deformation field at the level of different structural elements (Table 1).

**Table 1.** Stress and deformation fields at the level of different structural elements.

| Position/Opening (Max. ref. Load 160 kN) | Alpha Angle [°] | H1/V1 Component [kN] | H2/V2 Component [kN] | N/T [kN] Components | Sigma vonMises [MPa] | Resulting Displace-ment Max [mm] | Cfl. |
|---|---|---|---|---|---|---|---|
| Closed/8 m | 9.31 | 12.33/0.5 | 46.52/308.3 | 323.34/18.5 | 67.72 | 28.41 | 19.4 |
| Median/20 m | 23.72 | 89.15/27.1 | 2.85/263.8 | 508.24/32.2 | 120 | 52.36 | 10.9 |
| Iteration 4 23.32 m | 30 | 107.1/41.13 | -/236.96 | 531.2/52.14 | 148 | 77.54 | 8.25 |
| Position acc. 24 m | 31.15 | 110.3/44.13 | -/231.3 | 540/53 | 150 (160) | 77.93 | 8.18 |
| Iteration 1 24.56 m | 32.11 | 113.1/46.9 | -/226.3 | 542/52 | 152 (158.3) | 78.35 | 8.14 |
| Iteration 2 25.15 m | 33.15 | 116.1/50 | -/220.7 | 550/51 | 153 | 77.71 | 8.16 |
| Iteration 3 26.27 m | 35.15 | 121/55.3 | -/210 | 561/50.5 | 153 | 78.03 | 8.1 |
| Iteration 5 26.73 m | 36 | 123.07/57.75 | -/205.39 | 567/50.4 | 155 | 78.3 | 8.05 |
| Iteration 6 27.28 m | 37 | 125.63/60.92 | -/199.46 | 575.27/48.9 | 155 | 77.06 | 8.11 |
| Open/32 m | 44.4 | 146.4/95.4 | 4.5/138.45 | 727/18 | 130.1 | 46.35 | 10.57 |

For instance, the sequence of the working stages is given for the case of one of the final situations studied (angle between the vertical and the longitudinal axis of the main arm of 31.150 or the opening of the load of 24 m, the payload of 16,000 daN); the chosen working variant corresponds to an opening of value that is easy to impose in reality (through the position transducers of the crane), being the closest situation to the most disadvantageous case, according to the data below.

This working position was also taken into account for tensometric measurements made in situ, the presence of a set of such measurements (with all the additional elements imposed by the restrictive requirements of the method) being motivated by the need to correlate the data obtained by theoretical study (analytical calculation and numerical methods) with the reality.

The accuracy of the quantities handled at this level of the study was verified by comparison (graphic method vs. analytical method), with the angle sizes being estimated in the first approximation with the help of software tools (Evaluate/Measure tab), respectively finalized (confirmed) with digital protractor.

In order to find the middle-section loading scheme, the variation of the sectional efforts was drawn along the elements, knowing that the start/end sizes in the sectional effort diagrams represent the very components of the reaction forces (axial force and shear force, respectively—N, T components, see Table 1, Figures S6 and S7).

On the basis of the static study thus outlined, further steps can be taken with reference to aspects related to the loss of the stability of the elastic equilibrium form (buckling) or fatigue study.

According to the elementary relations in Mechanics of Materials [13], for the initial cross section in question, the geometrical characteristics are established, in this case the area moment of inertia and the section modulus, respectively, in relation to the neutral axis of the section (for efficiency, the calculation module of the geometric features within the AxisVM software (2013, Inter-CAD Software Development Company, Budapest, Hungary) suite was used).

The geometrical dimensions were taken from the original technical documentation of the machine (Figure S8). The calculation stage was also repeated for the new shape of the cross section, a shape that corresponds to the plating at the top of the box-shaped cross section (Figure S9); it is important to remember the pair of values for the geometrical characteristics mentioned in the new constructive variant. One can observe the significant variation of the values, which has as a main cause the change in the position of the centroid of the section, implicitly the neutral axis position.

Thus are obtained the area moments of inertia in the case of the original cross section (Figure S8) and in the case of the plated variant (Figure S9), respectively. Similarly, the values for the section modulus corresponding to both constructive variants are found, in this case:

$$W_2 = W_{y,el,t} = W_{y,el,b} = 6.26e6 \text{ mm}^3, \ W_2^{pl.} = W_{y,el,b}^{pl.} = 6.97e6 \text{ mm}^3; \tag{1}$$

The coincidence of values in the case of the original variant is caused by the double-symmetry character of the original section.

Taking into account only the term from the bending moment, there is an increase in the stiffness of the corrected (repaired) section, in the form of:

$$\frac{EI_2^{pl.}}{EI_2} = \frac{2.84e9}{1.94e9} \Rightarrow \frac{EI_2^{pl.}}{EI_2} = 1.46 \text{ (times)} \tag{2}$$

and an increase from a bending-capacity point of view of:

$$\frac{W_2^{pl.}}{W_2} = \frac{6.97e6}{6.26e6} \Rightarrow \frac{W_2^{pl.}}{W_2} = 1.11 \text{ (times)} \tag{3}$$

The problem becomes more intricate as one cannot speak of two separate structures, one of which is strengthened by plating along its entire length, but of a local deterioration for the technical (geometrical) parameters of the structure. In reality, plated areas are limited to a particular area of interest.

The experimental study had as its objective the determination of the stress field existent at the boom level of the crane subjected to mechanical or equivalent stresses (mass load in the hook). This was performed with the help of electroresistive tensometry, one of the most-used techniques for this kind of investigations.

The tensometric measurement stage consisted of placing the appropriate transducers in areas considered of main interest—their position can be observed in Supplementary File, Figure S6a–c—with the recorded strings of values (output data) being processed later.

Thus, in the lateral-middle area of the main section (halfway between the articulated extremity of the arm and the hinged ear for the counterweight), it was desired to mount three measuring points consisting of six plane-grid-type tensometric transducers laid in pairs at 900, in order to highlight normal tensions due to the horizontal bending of the arm—dynamic stresses occurred when moving it by turning in the area opposite to the main ear of the section, on its outer side, a measuring point of rectangular rosette-element type (three measurement grids superimposed at 00, 450 and 900, in order to assess the plane stress state due to the expected composed load).

If in the case of flat-grid transducers, the data processing is relatively easy, the readings in the recorder being multiplied directly with the coefficients of dimensional transformation and the longitudinal elasticity modulus of the material (steel, E = 2.1 × 10$^5$ MPa), in the case of the rosette transducer, the algorithm has two distinct calculation steps.

The determination of the main strains, as well as the position of the main axis system (angle), are made by the well-known relations within the Mechanics of Materials discipline, relations that have as input the registered strain measurements obtained by three-gauge tensometric rosettes (rectangular-type with 450 spaced grids).

Thus, noting with the values obtained by direct measurements, values corresponding to the positioning directions of the rosette-type transducer (see Figure 5), the main strains are obtained according to the conversion relations.

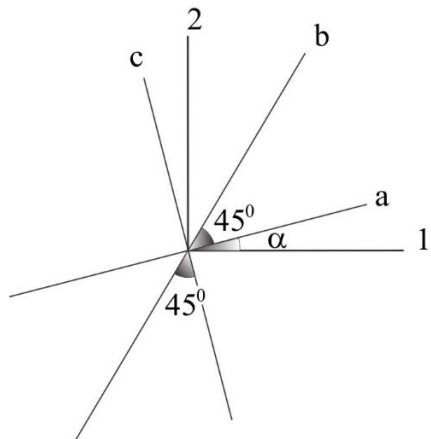

**Figure 5.** Main axis system position, where: 1,2 are cartesian reference system axis and a,b,c are axis of gauge tensometric rosette.

It should be noted that the verification of the intensity of the main stresses, as well as their directions, were performed with the specialized online computer of the manufacturer of the recording equipment (tensometric bridge), Vishay Precision Group, Malvern, PA, USA.

$$\varepsilon_{1,2} = \frac{\varepsilon_a + \varepsilon_c}{2} \pm \frac{1}{\sqrt{2}} \sqrt{(\varepsilon_a - \varepsilon_b)^2 + (\varepsilon_b - \varepsilon_c)^2}; \tag{4}$$

$$\alpha = \frac{1}{2} arctg \left( \frac{\varepsilon_a - 2\varepsilon_b + \varepsilon_c}{\varepsilon_a - \varepsilon_c} \right). \tag{5}$$

For a synthetic image of the phenomenon, the second processing step is to use the conversion formulas from main normal stresses to equivalent normal stress estimated according to the theory of maximum-deviation energy (failure criterion V or IV) or von Mises–Hencky criterion, in short. Starting from the fundamental general expression of the Vth failure criterion, in the form of

$$\sigma_{ech} = \sqrt{\sigma_1^2 + \sigma_2^2 + \sigma_3^2 - \sigma_1\sigma_2 - \sigma_2\sigma_3 - \sigma_3\sigma_1}, \tag{6}$$

in the case of the plane stress state, one can obtain

$$\sigma_{ech} = \sqrt{\sigma_1^2 + \sigma_2^2 - \sigma_1\sigma_2}, \tag{7}$$

or

$$\sigma_{ech} = \sqrt{\frac{1}{2}\left[(\sigma_1 - \sigma_2)^2 + \sigma_2^2 + \sigma_1^2\right]}. \tag{8}$$

## 3. Results and Discussion

As a result of the modeling, a static study was performed for the arm on which interventions were performed within the maintenance program, which has the following as output data:

- the equivalent von Mises stress criterion (equivalent normal stress expressed by the theory of the maximum potential energy of deviation (change of form)—(Figure 6) field, respectively;
- the deformation field given by the total resultant deformation parameter (Figure 7).

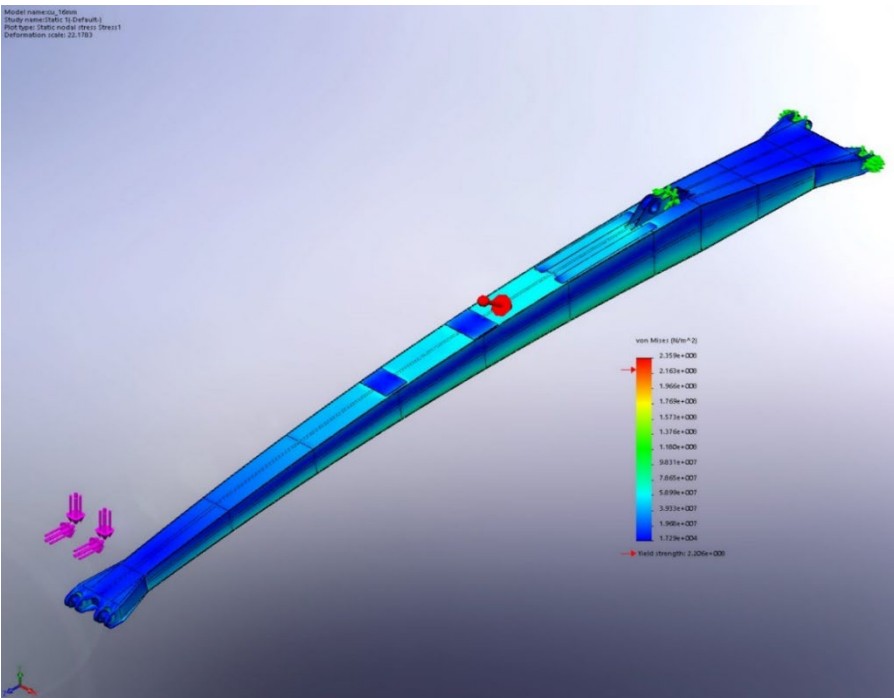

**Figure 6.** Equivalent normal stress—von Mises criterion.

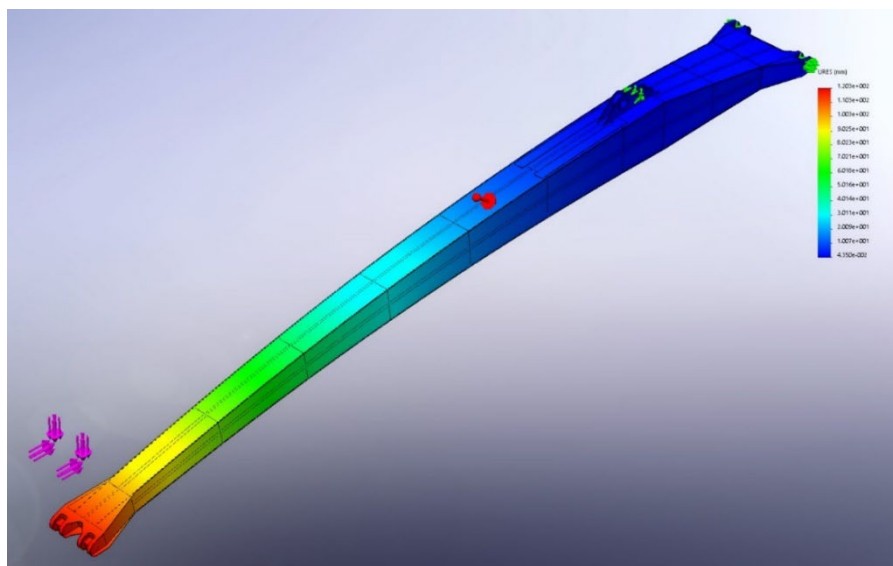

**Figure 7.** Deformation field.

It is worth mentioning that the first studies of this kind (from a SolidWorks solver point of view), were carried out with the following parameters: automatic iterative study; the "large assemblies" option activated; the "curvature based" meshing with implicit 136 mm element size; the h-adaptive method with three iteration loops; the final variants using a classic approach mode, with a much finer meshing algorithm (Figure 8) (element size of 30 mm, "curvature based mesh", increased solving time, important consumed resources), which allowed for the avoidance of the so-called singularity points, fictional regions with excessively accentuated stress-field values (Table 2).

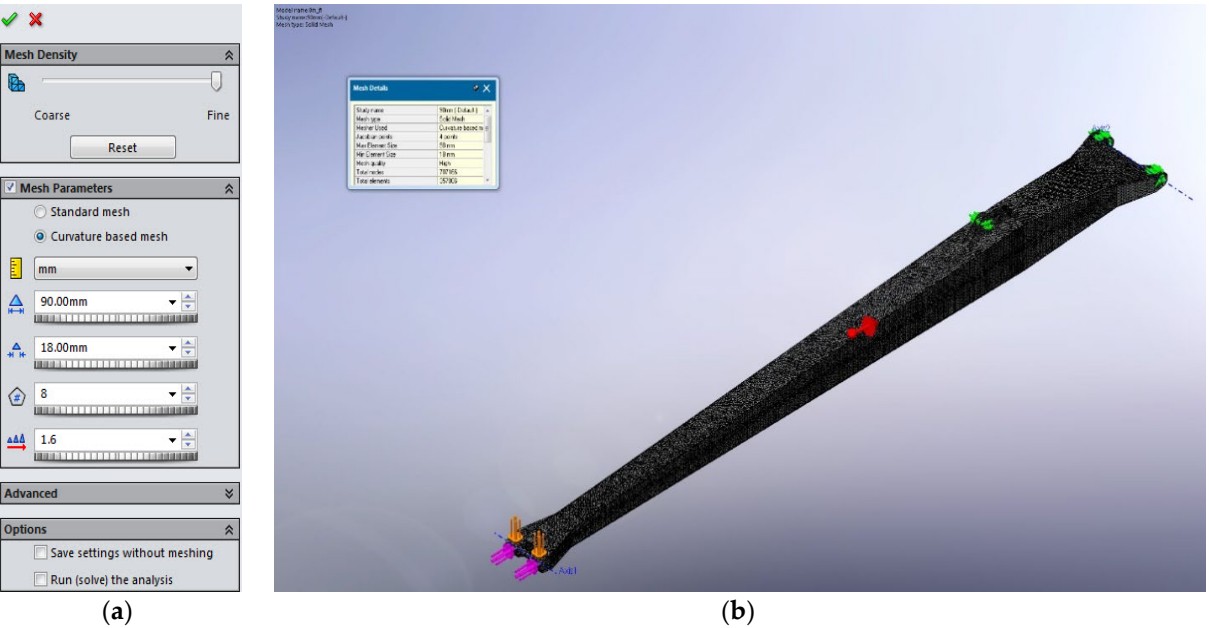

**Figure 8.** Main boom (arm) meshing (**a**) mesh parameters (**b**) mesh details.

**Table 2.** Comparison table from mesh element dimension point of view.

| Element Size [mm] | Degree of Freedom [D.O.F.] | No. of Nodes | No of Elements |
|---|---|---|---|
| 90 | 2,121,624 | 707,166 | 357,006 |
| 60 | 3,837,342 | 1,279,114 | 647,774 |
| 30 | 10,743,252 | 3,581,084 | 1,818,071 |

In addition to the static study, a Euler buckling study (Figure 9) was carried out on which occasion the areas prone to the occurrence of the phenomenon can be visualized, as well as the estimated buckling coefficient of safety, a criterion that also served to detect and confirm the so-called disadvantageous working position.

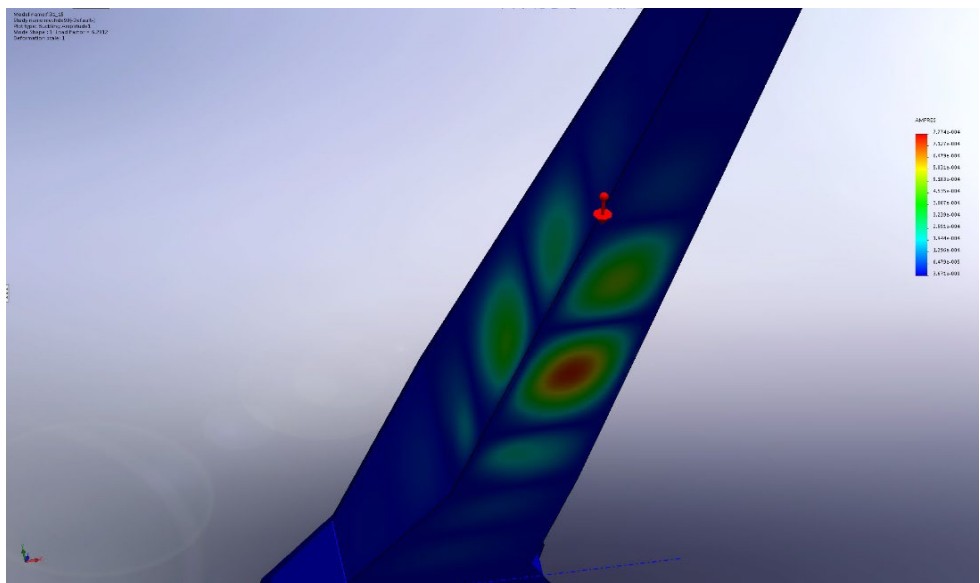

**Figure 9.** Buckling study.

To complete the overall picture, in some situations a stand-alone study of fatigue calculation was also run, which, depending on the initially permissible dataset, allows the estimation in terms of impairment (damage) or total life from a structural-behavior point of view (Figure 10). Fatigue analysis was based on existent facilities related to SolidWorks "Simulation" module. The used fatigue parameters and analysis conditions are:

- the calculation engine had as input size the previously conducted static study;
- the analysis conditions were those corresponding to the most unfavorable situation (maximum crane load with respect to boom position), specified above;
- the schematization used was "Soderberg";
- the thickness of the material was the lowest value determined by ultrasonic measurements (UTg).

As a result of the tensometric measurements and the postprocessing stage, according to those presented in the previous paragraph and the correlation of the data thus obtained for the dangerous section of the main section (the area of application of the rosette transducer), the obtained values are presented in Table 3, as follows:

**Table 3.** Data obtained for the dangerous section of the main section.

| Load [daN]/Position | Transducer Theoretical Value | Modulus Ecart [MPa] | Estimated Dynamic Coefficient |
|---|---|---|---|
| 16,000/ raised-lowered | 57.21 − 39.75 = 17.45 MPa | 16/17.8 | 1.4 |

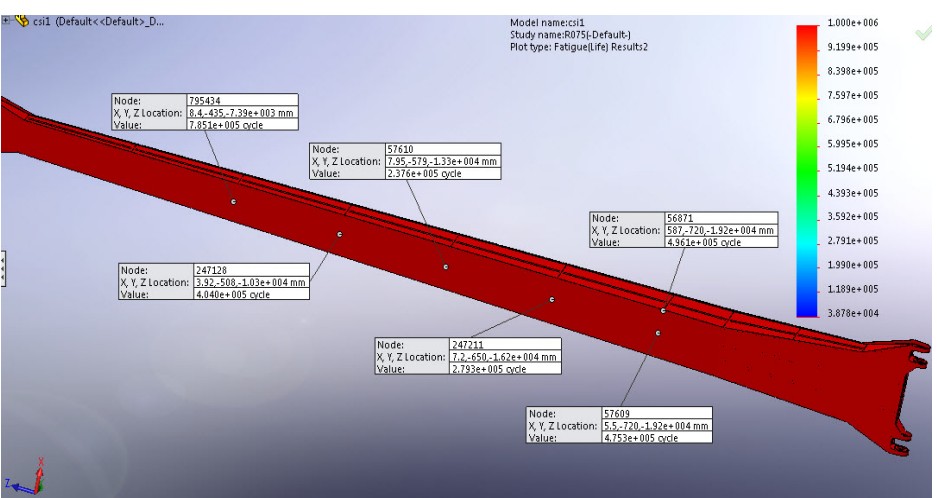

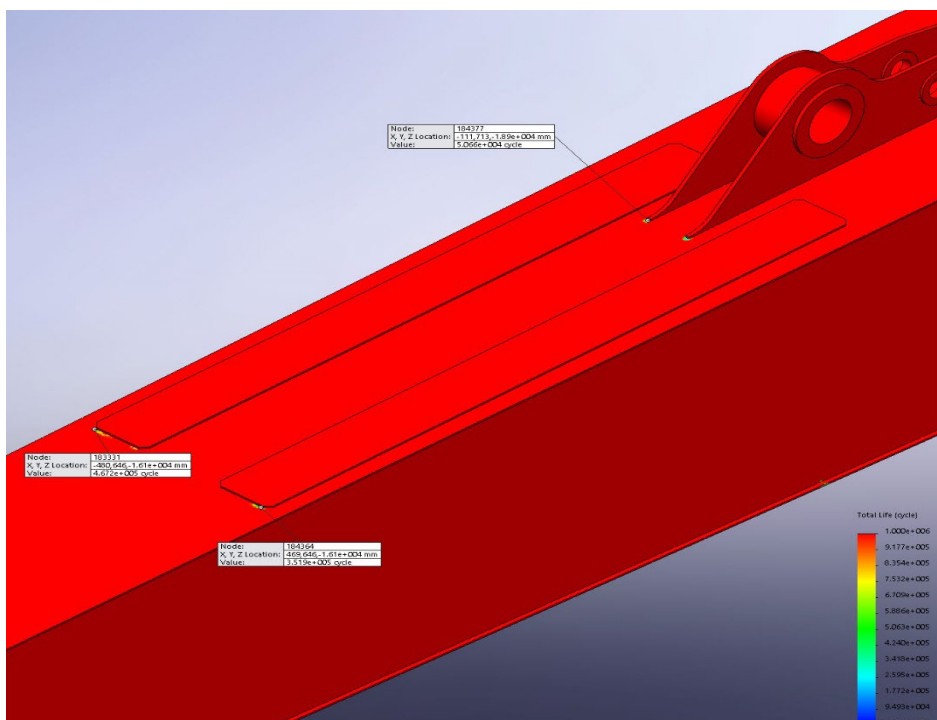

**Figure 10.** Fatigue study.

First of all, one should note the small weight of the value of the measured stresses (additional values due practically to the loads hung in the hook during the tests), taking into account the existence at the level of the structure of a state of tension prior to the positioning of the tensometric gauges, a stress state due to the structure's own weight. The existence of differences between the recorded values and the estimated theoretical ones, differences due to the significant dynamic effects of the (important) inertial masses in motion, were also found. Indeed, from the direct discussions with the beneficiary's representatives, including facts found directly during the operations of positioning the transducers and actually going

through the real measurement process, it is worth mentioning the clear presence of the dynamic component within the loading scheme of the structure. Thus:

- positioning of the load in the working range of 8 m–32 m, which is carried out by simultaneously operating all the components of the structure;
- the additional weight of the standard equipment of the studied crane is 7000 daN, which is permanently present but has been missing, for technical reasons, from the on-site tests (hydraulic control grapple);
- the existence of particularities concerning the actual way of using the equipment (assembly–disassembly of the grapple by its direct positioning on the ground—accidental shocks in case of partial failure;
- shock loads at the end of barge-unloading operations;
- the existence in the technical history of the machine of some periods of its use for the compaction of metal waste (also by shock), or other operations unsuitable for the purpose for which the machine was designed, etc.

The above comments require the broadening of discussion by recalling the elementary formulas and the relationships regarding the dynamic coefficients of impact in the case of shock loading; in the case of some elements that move at the "*v*" speed, the overload coefficient for shock-applied loads is expressed in the form of

$$\psi = 1 + \sqrt{1 + \frac{v^2}{g \cdot \Delta_{st}}},$$  (9)

in which:

*v*—the speed of evolution of the structural element;
*g*–gravitational acceleration;
$\Delta st$—the static displacement (cross-section slope), with respect to the applied load direction.

Even if one were to resort instead to simplified relations such as the study algorithm of the elevator cable (typical problem in Mechanics of Materials), writing the equilibrium equation in the form

$$F_i = \frac{Q}{g} \cdot a,$$  (10)

in which:

Be—the force of inertia due to the initiation of the motion under the action of an acceleration "*a*";
A—acceleration;
Q—payload;

and the cable axial load is

$$N = Q + F_i = Q\left(1 + \frac{a}{g}\right) \Rightarrow \psi = 1 + \frac{a}{g},$$  (11)

which is a situation that would lead to dynamic coefficients around 1.1–1.2. To the best of our knowledge, the prescribed values from the specialized literature, as well as the absence of elements dedicated to shock absorption (the inherent elasticity of the entire structure plays this role), place us around the value Ψ = 2, minimum value according to the relationship of the initial dynamic coefficient presented (the operating speed of the machine is, according to the data from the technical documentation, of 61 m/min, the designated coefficient having the value of 2.2).

The presence of the Δst deformation in static regime at the denominator of the second term below the radical (static deflection/slope), imposes the opening of a new field for debate, that of the opportunity to refurbish the structural components in question (introduction of additional gussets, increase in wall thickness, introduction of new additional box-shaped elements, etc.).

Not entirely excluding the need to carry out such operations in order to guarantee proper functioning and to extend the duration of use for the existing fleet (including the use of equipment coming from the so-called "second-hand" products market), it is recommended to study the technical consequences of such "repair" operations as thoroughly as possible, including an increase in weight of the structure (therefore increased inertial masses), simultaneously with a certain decrease in inherent elasticity terms (or increase in rigidity). For example, by decreasing the size of the values of the static deflections as a result of increasing rigidity, the dynamic coefficient of overload increases. Thus, by increasing the rigidity of the structure, its slenderness coefficient is reduced, with implications on increasing the danger of occurrence of the loss of stability for the elastic equilibrium form (buckling).

### 4. Conclusions

- A major influence concerning the stress- and deformation-field magnitude is due to the own weight of structural elements combined with dynamic phenomena related to "real-life" technological processes, with direct implications on service life.
- The nonlinear dependence between the stress-field mode of variation in the area of interest versus the load magnitude is noteworthy. We consider the existence of present nonlinearity due to tensometric strain-field assessment on the intermediate structural member of a deformable quadrilateral mechanism (the whole crane structure), the only pseudo-linearity being eventually expected at the top level of the mechanism itself (the so-called beak section).
- Taking into account the data obtained for the detailed main boom (arm) modelization, it is considered of interest to treat at least the upper beak-type section in the same manner, for effective highlighting of the particularities concerning its load-related behavior.
- The method of treatment by isolation of the structural components is motivated by the need to concentrate the calculation resources on a single element (possibly a group of elements), with direct implications on the degree of accuracy of the results. By comparison, the analysis conducted with a similar level of final data convergence for the entire assembly would involve superior computing resources in relation to the bulk of information obtained.
- Verification by tensometric measurements provides important information related to the behavior under load of the investigated structural element.

In order to increase the life span and safety operation of the main boom and implicitly of the entire crane, three major conclusions can be drawn:

1. It is recommended, in this particular case of study, to limit the lifting speed to a value of less than 60 m/ min, which would lead to an important decrease (over 25%) of the dynamic coefficient $\psi$;
2. It is recommended to strictly avoid the appearance of lateral loads;
3. Any attempt to "strengthen" the arm will require as thorough a study as possible of the technical consequences of such "refurbishment" operations, these generally resulting in an increase in weight of the entire structure (increased inertial masses), simultaneously with a certain decrease in its inherent elasticity (or increase in rigidity). For instance, by decreasing the size of the values of the static deflections as a result of increasing the rigidity, the dynamic coefficient of overload increases; otherwise, by increasing the rigidity of the structure, its slenderness coefficient is reduced, with implications on the increase in the danger of occurrence of the loss of stability of the elastic equilibrium form (buckling). As a result, it is recommended to replace the affected area as a result of environmental factors with material having a thickness equal to that of the original project, taking into account:
   - The fact that resulting geometry does not lead to excessive local stiffening of the section;

- The decrease in the probability of occurrence of tension concentrators—fragile breaking primers by methods that involve the gradual increase in wall thickness.
- The specific welding parameters (with postoperation treatment), in order to reduce the associated thermal influence.

**Supplementary Materials:** The following are available online at https://www.mdpi.com/article/10.3390/met12071107/s1, Figure S1. Archive documentation, Figure S2. Superior arm, Figure S3. Quadrilateral mechanism-closing elements, Figure S4. Counterweight, Figure S5. Main boom (arm) section, Figure S6. a–c Transducer-positioning scheme, Figure S7. Graphic approach, Figure S8. Analytical approach, Figure S9. Initial variant cross section, Figure S10. Plated variant cross section, Figure S11. Equivalent normal stress—von Mises criterion, Figure S12. Deformation field, Figure S13. Main boom (arm) meshing, Figure A14. Buckling study.

**Author Contributions:** Conceptualization, V.F. and L.R.; methodology, Ș.M. and V.F.; software, Ș.M.; validation, L.R. and V.F.; formal analysis, Ș.M.; investigation, V.F., L.R., C.R. and A.B.; resources, A.B.; data curation, Ș.M. and V.F.; writing—original draft preparation, V.F. and A.M.; writing—review and editing, A.B. and C.R.; visualization, Ș.M. and A.M.; supervision, L.R. All authors have read and agreed to the published version of the manuscript.

**Funding:** This research received no external funding.

**Institutional Review Board Statement:** Not applicable.

**Informed Consent Statement:** Not applicable.

**Data Availability Statement:** Not applicable.

**Conflicts of Interest:** The authors declare no conflict of interest.

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
