# Peer review of "Novel Procedures for Sustainable Design in Structural Rehabilitation on Oversized Metal Structures"

_metals, doi:10.3390/met12071107_

Round 1

Reviewer 1 Report

Comments to the Author

Title: Novel Procedures for Sustainable Design in Structural Rehabilitation on Oversized Metal Structures

Authors: Mocanu Stefan, Rece Laurentiu, Burlacu Adrian, Florescu Virgil, Corneliu Rontescu, Arina Modrea

Comments: The operational risks of large port lifting equipment are high. This article aims to identify risk factors and to improve the operational safety and service life of various equipment. After a combination of theoretical and experimental studies confirming the validity of the theoretical model used, this paper finally provides specific sustainable solutions as well as new evaluation methods that can extend the life of the equipment and improve the operational safety of the equipment. However, some results are not well interpreted, and there are some problems in the article. The article should be major revised according to the following comments:

1. The author uses “KANGUR” in the Abstract and “Kangur” in the Introduction. The writing format is not uniform.

2. In the Introduction, why does the author discuss the work of other researchers in subsections, and is there any basis? And the author basically describes only the methods used by these researchers, but not their findings.

3. The research is based on the application of the finite element method and related simulation capabilities, but its application background is not explained by the author.

4. The author verifies the simulation results through experimental research, but the results of the experimental research are rarely mentioned by the author in the article.

5. In figure 2, there are studied two constructive variants of the main arm of the equipment. Can the author give details to reflect the difference in the internal structure?

6. The conclusions drawn by the author are not detailed in the Results and Discussion section.

7. The figures in the article are not clear.

8. The article cites too few references. The latest papers which may be helpful for this article are suggested for citation: Atomistic insights into the deformation mechanism of a CoCrNi medium entropy alloy under nanoindentation; Design and Characterization of Self-Lubricating Refractory High Entropy Alloy-Based Multilayered Films; Enhancing the tribological performance of the TiZrHfCuBe high entropy bulk metallic glass by Sn addition; Design and characterization of metallic glass/graphene multilayer with excellent nanowear properties; Effects of structure relaxation and surface oxidation on nanoscopic wear behaviors of metallic glass.

Author Response

We are very grateful to the reviewer for their critical comments and thoughtful suggestions.   Based   on   these   comments   and  suggestions, we   have   made   carefully modifications to the original manuscript. The point-to-point replies and explanations for all of the revisions are listed below for easy reference. We hope that we have understood this time what changes are needed to increase the level of the paper.

Point 1. The author uses “KANGUR” in the Abstract and “Kangur” in the Introduction. The writing format is not uniform.

Response1. Changes have been made.

Point 2. In the Introduction, why does the author discuss the work of other researchers in subsections, and is there any basis? And the author basically describes only the methods used by these researchers, but not their findings.

Response 2. In section “Introduction”, the authors included references to emphasize the fact that FEM Analysis is a modern method also used by other authors. The results obtained by other authors interested the elaboration team only in the context of the validation of the method. A clarification of this idea was brought by the insertion of lines 87 - 91 in the "Introduction" section.

Point 3. The research is based on the application of the finite element method and related simulation capabilities, but its application background is not explained by the author.

Response 3. We consider SolidWorks Dassault theoretical background (handbook) as relevant for the FEM domain.Furthermore, we added two extra references related to FEM – SolidWorks correlation that we consider relevant, respectively [14], [15].

Point 4. The author verifies the simulation results through experimental research, but the results of the experimental research are rarely mentioned by the author in the article.

Response 4. The authors presented the relevant results;

Point 5. In figure 2, there are studied two constructive variants of the main arm of the equipment. Can the author give details to reflect the difference in the internal structure?

Response 5. Figure 2 has as reference the sets of drawings existing within the technical documentation. The models are distinguished by a completely redesigned shape of the diaphragms as well as another way of solidarizing the strengthening rails of the "flange-type" and "web-type" elements. These aspects are presented in lines 128 – 137;

Point 6. The conclusions drawn by the author are not detailed in the Results and Discussion section.

Response 6. The conclusions are a direct consequence of the "Results and Discussions" section;

Point 7. The figures in the article are not clear.

Response 7. Graphical representation improvements, with respect to software limitation capabilities, are included in Supplementary Materials;

Point 8. The article cites too few references. The latest papers which may be helpful for this article are suggested for citation: Atomistic insights into the deformation mechanism of a CoCrNi medium entropy alloy under nanoindentation; Design and Characterization of Self-Lubricating Refractory High Entropy Alloy-Based Multi-layered Films; Enhancing the tribological performance of the TiZrHfCuBe high entropy bulk metallic glass by Sn addition; Design and characterization of metallic glass/graphene multilayer with excellent nano wear properties; Effects of structure relaxation and surface oxidation on nanoscopic wear behaviours of metallic glass.

Response 8. Three bibliographic references have been added; section “Introduction” has been completed. It was specified that " A Molecular Dynamics (MD) Simulation analysis for the metallic alloy structure, as proposed by Dongpeng Hua et al. [13], will be the subject of further research.", lines 84 – 86. One of the authors (Prof. Florescu) has concerns in the field of biotribology and who used the method of multi-pass indentation (proposed by Xu), in the work “The abrasion resistance estimation of the C120 steel by a multi-pass dual indenter scratch test, Journal Tribology, 16, 30 – 41, (2018)” will try to contact the authors of the paper "Effects of structure relaxation and surface oxidation on nanoscopic wear behaviours of metallic glass" (cited in the present article), in order to try a collaboration.

Reviewer 2 Report

In this manuscript, the authors presented several studies and advanced research carried out on the subject of large KANGUR type port cranes. The risk factors and to offer sustainable solutions for increasing operational safety and service life for the respective equipment are investigated. The overall quality of the manuscript is good, the topic is worth investigating. However, I do have some suggestions/comments that could improve this paper as follows:

1.      The originality of the method used in this paper should be stated more clearly. It is suggested that the writing of the abstract should be revised highlighting the novelties.

2.      The difference and similarities between this work and other published papers must be discussed.

3.      The dimensions are small in figure 5, it should be improved.

4.      Is the implementation of the present work complicated? Which issues may cause challenging?

5.      To the extent of this research, are there any recommendations for future work?

Author Response

We are very grateful to the reviewer for their critical comments and thoughtful suggestions.   Based   on   these   comments   and   suggestions, we   have   made   careful   modifications to the original manuscript. The point-to-point replies and explanations for all of the revisions are listed below for easy reference. We hope that we have understood this time what changes are needed to increase the level of the paper.

Point 1.      The originality of the method used in this paper should be stated more clearly. It is suggested that the writing of the abstract should be revised highlighting the novelties.

Response 1. Section “Abstract” has been revised;

Point 2.      The difference and similarities between this work and other published papers must be discussed.

Response 2.  Clarification was brought by the insertion of lines 87 - 91 in the "Introduction" section.

Point 3.      The dimensions are small in figure 5, it should be improved.

Response 3. Graphical representation improvements, with respect to software limitation capabilities, are included in Supplementary Materials;

Point 4.      Is the implementation of the present work complicated? Which issues may cause challenging?

Response 4. These aspects are caught in the "Conclusions" section, being highlighted starting with line 378;

Point 5.      To the extent of this research, are there any recommendations for future work?

Response 5. The recommendations regarding the maintenance of the presented equipment are presented in the "Conclusions" section. The methodology approach presented in this article is suitable to be applied for any type of metallic structure requested, of course considering their specific design.

Reviewer 3 Report

The topic is interesting and deals with the design aspects that must be considered in real big metal structures. However, there are some major concerns/comments on this study.

Based on the paper's title, I expected a novel design method to be proposed for the structure under consideration, but the work shows in fact rough stress and fatigue life analysis of the structure using SolidWorks, and finally, it provides some recommendations for the design of such structures. So perhaps the title should be changed to better fits the content. For stress and fatigue analysis, it should be noted that SolidWorks does not provide accurate stress results and this data cannot be considered for safe design purposes. The section on fatigue analysis is incomplete and many more details on the used fatigue model and the life analysis technique should be given. Again, SolidWorks does not accurately estimate the fatigue life and the results cannot be incorporated into a safe design. Considering these points, the contribution of the work in the scientific field is limited. 

The text is difficult to read and the English should be improved.

The quality of the images is very low and the texts in the images are too small.

Annex is not available at www.mdpi.com/xxx/s1, please check.

Based on the above comments, the work needs major improvement, especially in the numerical part and especially the fatigue section. As such, I would not recommend the paper for publication in Metals. 

Author Response

We are grateful to the reviewer for his critical comments and opinions, therefore the author’s response is as stated above:

The choise of using the D’Assault SolidWorks suite are based on:

  • It is a well known fact that a large scientific community members reports their study results based on FEM analysis, are using the suite of programs mentioned and frequently validated their theorical analysis by experimental methods.

 As a matter of fact, some co-authors published similar studies in Sensors Journal, in late 2020. Also, some co-authors currently have reviewed such articles. The authors have referred to this software in the revised form of  this article, in accordance with the requirements of the other reviewers;

  • The suite allows the creation of 3D models with very good accuracy because it contains a specific design section;
  • The analysis regarding fatigue calculus engine and implicitly the estimation of the remaining life of the equipment with high risk is applied by one of the co-authors in the professional expertises, these being validated by the National Authority. The method is approved by the Authority, validating the results of the analysis performed. In fact, this suite is the only one approved by the Regulatory Authority.

Regarding the quality of the images, graphical representation improvements, with respect to software limitation capabilities, are included in Supplementary Materials.

The attachment containing images related to the article has been enriched and reloaded.

The authors are always available with numerous examples in support of the above.

All authors state that they have no interest in the developers or distributors of the software suite.

Round 2

Reviewer 1 Report

The manuscript has been improved to a high level. It can be accepted now.

Author Response

Thank you very much for your valuable comments!

Reviewer 3 Report

Figures 6 to 10: Increase the font size of the texts shown on the images. 

Fatigue part: give enough information in the text about the fatigue analysis part, including a detailed description of the fatigue model, parameters used in the fatigue model, the way they have been obtained, and also the life analysis calculations were carried out. 

Author Response

We thank the reviewer for the relevant observation.

Point 1: Figures 6 to 10: Increase the font size of the texts shown on the images.

Response 1: Graphical representation improvments, with respect to software limitation capabilities, are included in the enhanced Supplementary Materials. The pictures in the appendix have the facility to be enlarged as much as necessary by the reader, the text remaining visible.

Point 2: Fatigue part: give enough information in the text about the fatigue analysis part, including a detailed description of the fatigue model, parameters used in the fatigue model, the way they have been obtained, and also the life analysis calculations were carried out.

Response 2: Clarification was brought by the insertion of lines 284 - 291 in the "Results and Discussion" section.